# Extended receptor repertoire of an adenovirus associated with human obesity

A. Manuel Liaci[1], Naresh Chandra[2], Sharvani Munender Vodnala[2], Michael Strebl[1], Pravin Kumar[2], Vanessa Pfenning[1], Paul Bachmann[1], Rémi Caraballo[3,4], Wengang Chai[5], Emil Johansson[3,4], Mikael Elofsson[3,4], Ten Feizi[5], Yan Liu[5☯*], Thilo Stehle[1,6☯*], Niklas Arnberg[2,3☯*]

1 Interfaculty Institute of Biochemistry, University of Tuebingen: Eberhard Karls Universitat Tubingen, Tuebingen, Germany, 2 Department of Clinical Microbiology, Umeå University, Umeå, Sweden, 3 Umeå Centre for Microbial Research, Umeå University, Umeå, Sweden, 4 Department of Chemistry, Umeå University, Umeå, Sweden, 5 Glycosciences Laboratory, Department of Metabolism, Digestion and Reproduction, Imperial College London, London, United Kingdom, 6 Department of Pediatrics, Vanderbilt University School of Medicine, Nashville, Tennessee, United States of America

☯ These authors contributed equally to this work.
* yan.liu2@imperial.ac.uk (YL); thilo.stehle@uni-tuebingen.de (TS); niklas.arnberg@umu.se (NA)

**Data Availability Statement:** All data needed to evaluate the conclusions in the paper are present in the paper and/or the Supplementary Materials. The following structures have been submitted for

## Abstract

Human adenovirus type 36 (HAdV-D36) has been putatively linked to obesity in animals and has been associated with obesity in humans in some but not all studies. Despite extensive epidemiological research there is limited information about its receptor profile. We investigated the receptor portfolio of HAdV-D36 using a combined structural biology and virology approach. The HAdV-D36 fiber knob domain (FK), which mediates the primary attachment of many HAdVs to host cells, has a significantly elongated DG loop that alters known binding interfaces for established adenovirus receptors such as the coxsackie- and adenovirus receptor (CAR) and CD46. Our data suggest that HAdV-D36 attaches to host cells using a versatile receptor pool comprising sialic acid-containing glycans and CAR. Sialic acids are recognized at the same binding site used by other HAdVs of species D such as HAdV-D37. Using glycan microarrays, we demonstrate that HAdV-D36 displays a binding preference for glycans containing a rare sialic acid variant, 4-*O*,5-*N*-diacetylneuraminic acid, over the more common 5-*N*-acetylneuraminic acid. To date, this sialic acid variant has not been detected in humans, although it can be synthesized by various animal species, including a range of domestic and livestock animals. Taken together, our results indicate that HAdV-D36 has evolved to recognize a specialized set of primary attachment receptors that are different from known HAdV types and coincides with a unique host range and pathogenicity profile.

## Author summary

Most human adenoviruses do not infect animals. HAdV-D36 stands out, as it can infect a wide range of animals and cause obesity in them. It is also associated with obesity in humans. Here, we demonstrate that HAdV-D36 can use a diacetylated sialic acid monosaccharide (4-*O*-acetyl-Neu5Ac) as a cellular receptor. 4-*O*-acetyl-Neu5Ac is synthesized

deposition in the Protein Data Bank (PDB): 9FAE (HAdV-D36 in complex with 4-O-acetyl-3'-sialyllactose), 9FAF (HAdV-D36 in complex with Neu4,5Ac2), 9FAG (HAdV-D36 in complex with Neu4,5,9Ac3), and 9FAH (HAdV-D37 in complex with Neu4,5Ac2).

**Funding:** This work has been funded by the SFB 685 of the German Research Foundation (TS), the Swedish Research Council (2013-2753 and 2013-8616, NA), Knut & Alice Wallenberg Foundation (KAW 2013.0019, NA), and The Swedish Cancer Society (CAN 2011/340, NA). The glycan microarray studies were performed in the Carbohydrate Microarray Facility at the Imperial College Glycosciences Laboratory supported by the Wellcome Trust Biomedical Resource Grants (WT099197/Z/12/Z, 108430/Z/15/Z and 218304/Z/19/Z) and in part by the March of Dimes Prematurity Research Center grant 22-FY18-82. The funders had no role in study design, data collection, and analysis, decision to publish, and preparation of the manuscript.

**Competing interests:** The authors have declared that no competing interests exist.

in animal cells but not in human cells. In humans, 4-*O*-acetyl-Neu5Ac is expected to be metabolized upon dietary intake and presented on cells in similarity with other non-human sialic acids. We conclude that HAdV-D36 has evolved to recognize a distinct receptor that is different from those used by other human adenoviruses.

## Introduction

Human adenoviruses (HAdVs) are nonenveloped, double-stranded DNA viruses that cause diseases in eyes, gut, and airways [1,2]. Species D HAdVs (HAdV-Ds) are investigated or used as vaccine vectors to protect against diseases caused by severe acute respiratory syndrome coronavirus 2 [3,4], Ebola virus [5,6], and respiratory syncytial virus [7]. Human adenovirus type 36 (HAdV-D36) is a member of HAdV species D and was first isolated in 1983 from a gastroenteritis patient [8]. HAdV-D36 was found to be serologically unique, which prompted its use as an experimental replacement for SMAM-1, an avian adenovirus that causes obesity in infected animals [9,10]. Interestingly, HAdV-D36 was shown to infect chickens, rhesus monkeys, marmosets, mice, and rats alike, causing obesity in all of them [9,11–13]. Moreover, the virus was shown to be transmitted horizontally from infected to healthy chickens through natural routes [12]. Both findings were surprising, as most HAdVs do not infect other species by natural routes and predominantly infect mucosal epithelial tissues. The link between HAdV-D36 infection and seropositivity with human obesity has been investigated at large scale (reviewed in [13–26]). However, the role of HAdV-D36 infection in human adiposity is still controversially discussed [27]. Host cell attachment is mediated by a variety of receptor molecules. Elucidating the molecular basis of these interactions is crucial for determining viral tropism, treatment of diseases, and potential therapeutic applications (reviewed in [28]). HAdVs use several cellular receptors including the coxsackievirus and adenovirus receptor CAR (species A, C-G types) [29–31], CD46 (B and D types) [32–34], desmoglein 2 (B types) [35], and sialic acid-containing glycans (D types) [36,37]. These interactions are all mediated by the C-terminal knob domain of the fiber capsid proteins, except for D types that engage CD46 via the hexon protein. Most HAdVs carry highly conserved RGD motifs in the penton base proteins that facilitate cell entry through interactions with cellular integrins [38,39]. HAdVs have been developed and used as vectors for multiple applications. Vectors based on, for example, HAdV-C5 target hepatocytes when administered intravenously. This extended, off-target tropism is mediated by coagulation factor X (FX) that bind to the HAdV-C5 hexon, and the HAdV:FX complex binds to heparan sulphate (HS) on the surface of hepatocytes [40]. Other vector types bind to other coagulation factors and lactoferrin, which is also associated with extended tropism [41–43]. HAdV-D types engage similarly a group of diverse receptors such as CAR, CD46, and sialic acid-containing glycans on cell surfaces. Use of sialic acid as primary attachment receptor has been shown for several HAdV-D types such as HAdV-D8, 19p, 37, and 64 [44–48]. The HAdV-D26 has been shown to use sialic acid as cellular entry receptor [49].

Sialic acids, in the animal kingdom, occur in the form of 5-N-acetylneuraminic acid (Neu5Ac), 5-N-glycolylneuraminic acid (Neu5Gc), and 2-keto-3-deoxy-D-glycero-D-galacto-nononic acid (KDN) depending on the animal species [50]. Neu5Ac is the most commonly expressed form of sialic acid in mammals, including humans. It often undergoes various post-glycosylation modifications including O-acetylation. The 7-O-, 8-O-, and 9-O-acetylated forms of Neu5Ac are broadly found across animal species. In contrast, 4-O-acetylated Neu5Ac is expressed in fewer species and has so far not been detected in humans [50–52]. In this manuscript, we will use the term SA for sialic acid in general and Neu5Ac specifically as

appropriate. The O-acetylated form of Neu5Ac will be abbreviated, for example 4-O-acetyl-Neu5Ac to Neu4,5Ac$_2$, 9-O-acetyl-Neu5Ac to Neu5,9Ac$_2$. It is interesting to notice that several of these SA-variants are present in domestic animals. Such host reservoir may indicate a route of virus transmission through contact with domestic animals or the consumption of livestock, respectively. Despite the extensive research that has been conducted on HAdVs, information about the determinants of HAdV-D36 unusual tropism and its ability to horizontally spread between animals has remained elusive to date. This prompted us to investigate the primary attachment receptor portfolio of the virus, which is believed to be one of the determinants of HAdV tropism. We have employed a combined structural biology and virology approach. We postulate that the virus engages one or several unique receptors that might at least partially account for the observed phenomena.

## Results

### HAdV-D36 depends on CAR and SA-containing glycoproteins for attachment to A549 cells

In order to assess the attachment efficiency of HAdV-D36 to cells, we first searched for a suitable model cell line. We analyzed the binding of $^{35}$S-labeled HAdV-D36 to a range of cell lines originating from different tissues such as lung, small intestine, liver, and kidney. All cell lines displayed similar binding profiles for HAdV-D36 (**S1A Fig**). Given its adipogenic potential, we also compared the virus binding to A549 cells and cell lines originating from adipose tissue. HAdV-D36 bound slightly better to A549 cells (**S1B Fig**). Thus, A549 cells were used for characterizing the nature of the cellular receptor used by HAdV-D36. To this end, we analyzed HAdV-D36 binding to enzymatically and chemically pretreated A549 cells. Pretreatment of cells with the proteases ficin (**Fig 1A**) and proteinase K (**S2A Fig**) significantly reduced the attachment of $^{35}$S-labeled HAdV-D36 virions to cells. In contrast, proteases such as bromelain and V8 did not have any effect (**S2B–S2C Fig**). Treatment of cells with Benzyl-GalNAc, an inhibitor of de novo *O*-glycosylation, also reduced significantly the binding of the virus to cells (**Fig 1B**). Pre-treatment of cells with tunicamycin, which inhibits *N*-glycosylation, caused non-significant effect on virus binding to cells (**S2D Fig**). Treatment with P4 (DL-threo-1-phenyl-2-palmitoylamino-3-pyrrolidino-1-propanol, an inhibitor of glycolipid synthesis) and heparinase III (which removes cell-surface HS) (**S2E** and **S2F Fig**), did not alter virus binding to cells. Pre-incubation of viruses with soluble heparin, which is often used as a soluble analogue of HS only decreased binding of HAdV-D37, but not of HAdV-D36 (**S2G Fig**). Similarly, coagulation factors such as factor X did not affect the binding of HAdV-D36, but enhanced HAdV-C5 binding (**S2H Fig**) as predicted. Collectively, these data clearly indicate that surface molecules mediating the attachment of HAdV-D36 virions to A549 are either protein and/or glycoprotein entities, but not glycolipids or HS, and, that factor X is not involved in HAdV-D36 cell binding. Further, we screened for possible involvement of known HAdV attachment receptors by analyzing fiber knob (FK) binding to a library of Chinese hamster ovary (CHO) cells that overexpress the coxsackie- and adenovirus receptor (CAR), the complement regulatory protein CD46, and a number of known viral cell-surface receptors such as CD21, ICAM-1, and CD55, including control cell lines (CHO-K1 and CHO-mock) lacking these (**Fig 1C**). The data revealed substantially better attachment of all FKs investigated to CHO cells that ectopically express CAR, as compared to CHO-mock cells. We also observed lower FK binding of HAdV-D36 and -D37 to SA-lacking Lec2 cells as compared to the SA-expressing Pro5 cell line (parental to Lec2). These data suggest that CAR and SA-containing molecules contribute to the initial attachment of HAdV-D36 to these cells. Treatment of A549 cells with *V. cholerae* neuraminidase, which removes SAs from the cell surface, significantly

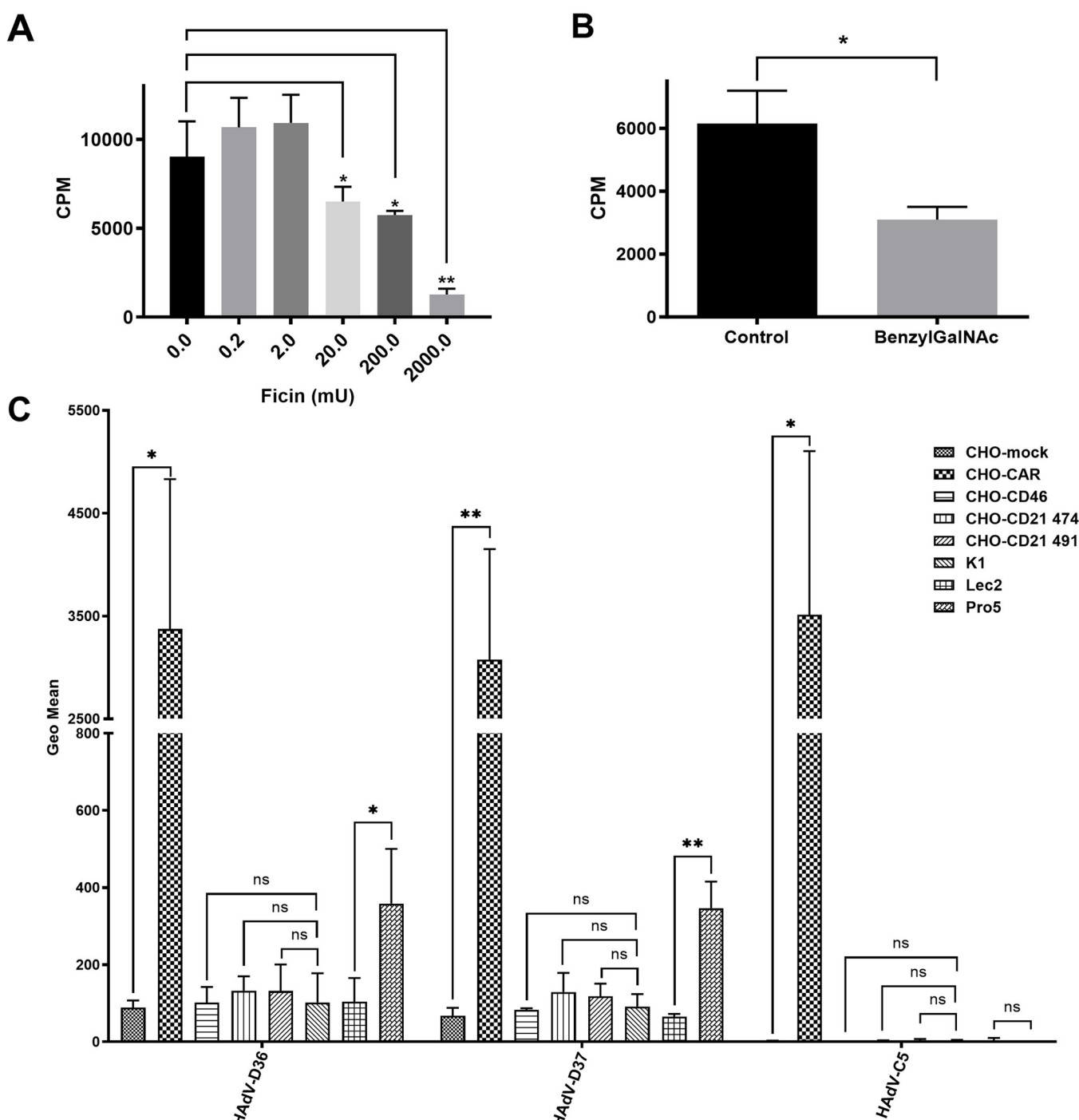

**Fig 1. HAdV-D36 uses CAR and sialic-acid-containing glycotopes to attach to A549 cells via the fiber knob.** (A) Binding of [35]S-labelled HAdV-D36 virions to A549 cells pretreated with different concentrations of the protease ficin. (B) Binding of [35]S-labelled HAdV-D36 virions to A549 cells cultured in the presence and absence of the metabolic inhibitor BenzylGalNAc (inhibitor of O-linked glycan synthesis). [35]S-labelled virion binding to cell was shown as CPM (counts per minute) in both A and B. (C) Binding of [35]S-labelled HAdV-D36 FK, HAdV-D37 FK, and HAdV-C5 FK to CHO cell lines expressing different human cell surface proteins (CAR, CD46, CD21; see the legend on the right), or overexpressing SA (Pro5). The CHO-K1 and CHO mock cells represent- control cell lines. Y-axis shows knob binding to cells, represented as Geo Mean (Geometrical mean). All experiments were performed three times with duplicate samples. Error bars represent mean ± SD. The statistical significance was determined by unpaired two-tailed t test; ns = not significant, * P of < 0.05 and ** P of < 0.01.

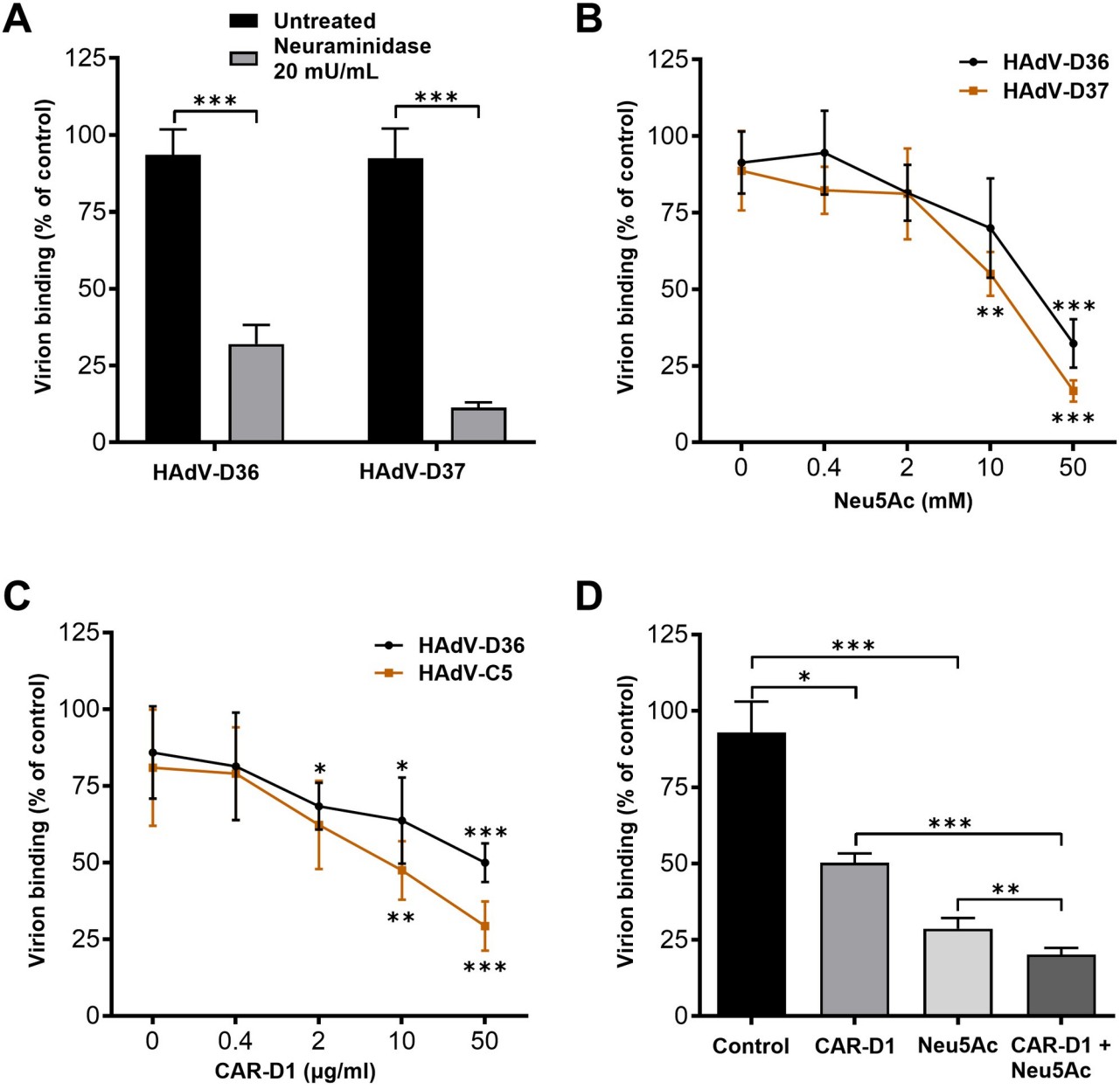

**Fig 2. CAR and Neu5Ac engagement have a synergistic effect on virion attachment.** (*A*) $^{35}$S-labelled HAdV-D36 and -D37 virion binding to A549 cells, treated with (black bars) or without (white bars) neuraminidase. (*B*) Binding of $^{35}$S-labelled HAdV-D36 and -D37 virion, pre-incubated with increasing concentration of Neu5Ac, to A549 cells. (*C*) $^{35}$S-labelled HAdV-D36 and -C5 virion binding to A549 cells. Virions were preincubated with increasing concentrations of soluble CAR-D1 before binding. (*D*) $^{35}$S-labelled HAdV-D36 virion binding to A549 cells in absence (control) or presence of CAR-D1 (50 μg/mL), Neu5Ac (50 mM), or both. All data are presented as % of control, where control for (A) refers to binding of virions to cells not treated with Neuraminidase and for (B-D) refers to binding of virions to cells where virions were not incubated with Neu5Ac or CAR-D1 or both before binding. All experiments were performed three times with duplicate samples. Error bars represent mean ± SD. The statistical significance was determined by unpaired two-tailed t test. * P of < 0.05, ** P of < 0.01 and *** P of < 0.001.

reduced (by ~70%) the attachment of HAdV-D36 (**Fig 2A**). As predicted, neuraminidase treatment also reduced (by ~85%) HAdV-D37 binding to cells. Furthermore, pre-incubation of virions with soluble Neu5Ac inhibited virion binding to cells in a dose-dependent manner (**Fig 2B**). Pre-incubation with the soluble domain 1 of CAR (CAR-D1) decreased HAdV-D36

binding to cells, albeit less efficiently than the binding of CAR to HAdV-C5 (**Fig 2C**). The canonical binding site for CAR and the SA-binding site of other species D HAdVs are located on distinct and spatially separated locations of the knob domain. Therefore, CAR and SA engagement are unlikely to interfere with each other. Indeed, pre-incubation of virions with both CAR-D1 and SA had a synergistic inhibitory effect on virion binding (**Fig 2D**).

## Structure of the HAdV-D36 FK and implications for receptor engagement

We determined the crystal structure of the HAdV-D36 FK to identify regions that usually mediate binding to SA, CAR, and other known receptors such as CD46. The crystal structure revealed that like all HAdV FKs, the HAdV-D36 FK is a compact trimer. Each monomer contains a β-sandwich domain, in which the β-strands of the two β-sheets (ABCJ and DIHG, respectively) are connected by long, surface-exposed loops. The HAdV-D36 FK has three major distinctive features when compared to other species D FKs. First, it has a reduced positive surface charge in comparison to HAdV-D37 (**Fig 3A**) [47]. Second, the HAdV-D36 FK has a narrower central cavity compared to the two other knobs (**Fig 3B**) [47]. The reason for this is an altered relative arrangement of the monomers in the trimeric knobs (RMSD = 0.79 Å) rather than a significant difference from the monomers of each type (RMSD = 0.48 Å). Third, HAdV-D36 possesses a markedly extended DG loop located at its sides (**Fig 3C**). With a total length of 21 amino acids, this loop is among the largest DG loops found in HAdV FKs.

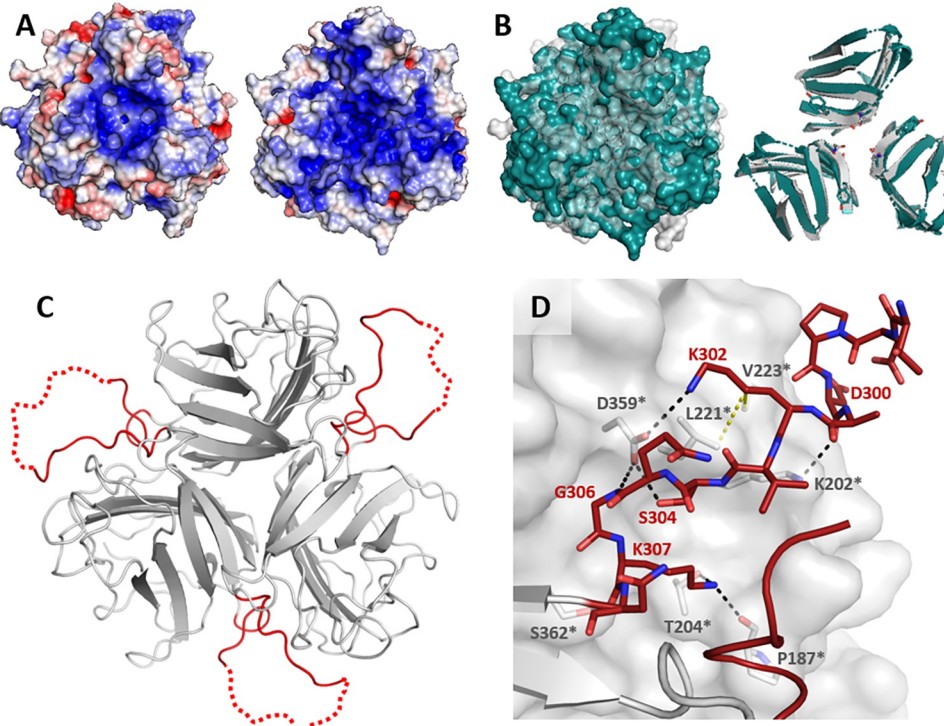

**Fig 3. Structural features of the HAdV-D36 FK.** (*A*) Poisson-Boltzmann electrostatic potential surface representation of the HAdV-D36 FK (left) and HAdV-D37 FK (right) calculated at ±3 kT/e and shown from a top view along the threefold axis. (*B*) Comparison of the quaternary arrangement of the FKs of HAdV-D36 (gray) and HAdV-D37 (green, PDB-ID 1UXE) displayed as surfaces and as ribbon representation of the beta sheet cores. The superposition was calculated using all atoms of both trimers. (*C*) The HAdV-D36 FK possesses long DG loops (red) that are partially ordered and contact the counterclockwise neighboring monomer when viewed from the top. (*D*) Close-up view of the interactions between the DG loop and the counterclockwise neighboring monomer. Polar and hydrophobic interactions are displayed as black and yellow dashes, respectively.

The loop is only partially flexible and has several interactions with the counterclockwise neighboring monomer involving a stretch of nine residues (**Fig 3C** and **3D**), probably contributing to the extraordinary stability of the FK (**S3 Fig**). Interactions are formed between the backbone of D300 and the side chain of K202 (AB loop). Additional interactions near the sheet are mediated by S304, Q305, G306 (all with D359) and K307 (with P187/T204, both AB loop). The bulky K302 forms a salt bridge with D359, while the alkane part of its side chain interacts with V223 and L221. Even at a pH below 3, the FK displays remarkable stability (melting point (Tm) 62.4˚C), which might be one of the prerequisites for its putative enteric tropism. The length of the DG loop is one of the determinants for binding of HAdV FK to important receptors such as CAR, and CD46 (reviewed in [53]). Structural superposition of the HAdV-D36 FK with HAdV-A12 FK in the CAR complex structure (PDB-ID 1KAC) (**S4A Fig**), and with HAdV-B11 in the CD46 complex structure (PDB-ID 2O39) (**S4B Fig**) reveal the proximity of DG loop with the canonical binding interfaces of both the CAR and the CD46. The main part of the HAdV-D36 canonical binding site shows a high shape complementarity to CAR (0.531 vs 0.793 in HAdV-D37, assessed by the SC program in CCP4 without considering side chain flexibility [54] (**S4C** and **S4D Fig**). The shown heavy clash between the DG loop of the counterclockwise adjacent FK monomer and CAR could only be overcome by a major rearrangement in the loop involving the dissociation of the inter-monomer contacts, or by altering the binding region. The HAdV-D36 FK readily forms a stable complex with the distal extracellular domain of CAR (CAR-D1) *in vitro*, similarly to the formation of the HAdV-D37/CAR-D1 complex (**S4E Fig**) [55]. On the other hand, the DG loop is predicted to interfere with a potential CD46 binding site, although no direct clashes were observed in the model (**S4B Fig**). The major determinants of CD46 binding that have been determined for HAdV-B11, B07, and B14 FKs are not conserved in HAdV-D36 [56]. HAdV-D48, which has a DG loop of similar length, cannot bind to CD46 [57]. In contrast to CAR, CD46 does not form a stable complex with the HAdV-D36 FK *in vitro* (**S4F Fig**). However, a recent study showed that HAdV-D26 and D56 binds to CD46 via the hexon protein [34], which added important information about additional mode of HAdV-D type interactions with CD46. Several other HAdV-D types including HAdV-D37 have been reported to use CD46 as a cellular receptor [34,58–63].

Alignment of the FK sequences of all species D serotypes (including HAdV-D8 to -D56) reveals two major clades of similar size with respect to the DG loop sequence (**S5A Fig**). The 18 members of the 'HAdV-D36-like' clade possess similarly elongated DG (or FG) loops and a VSN or VTN interface, while the 17 knobs of the 'HAdV-D37-like' clade containing all EKC-causing HAdVs have DG/FG loops shorter by eight to eleven residues and a YGT or YGN interface configuration. HAdV-D15 could not be assigned to any of the two clades. Visual analysis of the trimer interfaces of the HAdV-D36 and -D37 FKs revealed a stretch of three amino acids within the G-strand (*VSN*311-313 in HAdV-D36 and *YGT*308-310 in HAdV-D37) that likely determine the relative tilt of the monomers with respect to the trimeric axis of the FK, thus accounting for the narrower trimer interface in HAdV-D36 (**S5B** and **S5C Fig**). This region harbors the SA-binding site of HAdV-D37. The structure of the HAdV-D36 FK in complex with the synthetic SA analogue α-2-*O*-methyl-Neu5Ac corroborates the binding of Neu5Ac at the canonical SA-binding site (**S5D Fig**). In the case of the HAdV-D37 fiber knob, the two key polar interactions occur between residue K345 and the carboxyl group of Neu5Ac and between the backbone of P317 and the sugar's amide nitrogen, respectively (**S5E Fig**). Both interactions are formed with the β-face of the sugar and are also present in the HAdV-D36 complex structure, although the analogous position of P317 is occupied by a valine (V320) in HAdV-D36. The narrower binding cavity has several implications for the way HAdV-D36 engages Neu5Ac. On the one hand, this arrangement allows HAdV-D36 to contact the sugar from the α-face, using a second monomer to form a direct hydrogen bond

between N313 and the *N*-acetyl group of Neu5Ac, as well as a water-mediated contact between the backbone of W348 and the sugar's glycerol chain. HAdV-D37, on the other hand, mediates only water-bridged contacts to the α-face. Secondly, the binding cavity for the N-acetyl group is rearranged and becomes less spacious in the HAdV-D36 FK (**Figs 3A** and **S5D–S5F**). The residues forming this cavity are Y315 as well as V311 and F325 from the monomer facing the α-side of the sugar. In HAdV-D37, the *N*-acetyl group is accommodated in a canyon-shaped cavity formed mostly by two tyrosines (Y312 and Y308) from the neighboring monomer, as well as V322, which occupies the position analogous to F325. Overall, the HAdV-D36 FK quaternary structure causes a shift of the sugar in the direction of the α-face (**S5F Fig**). Upon superimposing monomers of the HAdV-D36 and -D37 fiber knobs (**S6A Fig**), it becomes apparent that this shift is not caused by differences in the tertiary structure. Despite the overall similarity of the binding, our attachment inhibition, and hemagglutination data consistently show that the HAdV-D36 binding to Neu5Ac is weaker compared to the binding of HAdV-D37 to Neu5Ac (**Figs 2B, S7A** and **S7B**). This discrepancy is likely to originate at least partially from the lower electropositive charge of the HAdV-D36 binding pocket compared to HAdV-D37.

## HAdV-D36 preferentially binds to a rare SA variant

We employed glycan array screening with a library of about 500 glycans to identify SA-containing glycans that support efficient binding of the HAdV-D36 FK (**Fig 4A** and **S1 Table**). Strikingly, HAdV-D36 FK showed little binding to the sialylated glycans in the array except for a 3'sialyllactose (3'SL) probe containing a rare SA variant with an additional 4-*O* acetylation (Neu4,5Ac$_2$; position 46 in the array, and referred to below as 4-*O*-Ac-3'SL), which displayed a relatively strong fluorescence signal. Glycan probes containing a 3'SL motif capped with the unmodified Neu5Ac showed little or no binding. In contrast, HAdV-D37 showed preferential binding to 3'SL over 4-*O*-Ac-3'SL and bound to several other sialyl glycan probes in the array (**S8 Fig**). This provided additional support for the conclusion that HAdV-D36 FK binds weaker to Neu5Ac than HAdV-D37 fiber knob. A second array that specifically probed for the binding to differentially acetylated 3'SL variants confirmed that HAdV-D36 FK binds to Neu4,5Ac$_2$ (**Fig 4B**). The gene conferring 4-*O*-acetylation of Neu5Ac in vertebrates has not been characterized to date [64–66]. Only in a few studies has 4-*O*-acetylated SA been detected in humans [67,68], and Neu4,5Ac$_2$ was not among them. However, Neu4,5Ac$_2$ is readily detected in a wide range of domesticated animals such as chickens, mice, rats, rabbits, guinea pigs, or horses, as well as many fish species including salmon, cod, herring, and trout. The 4-*O*-Ac-3'SL, the specific Neu4,5Ac$_2$-containing glycan has been reported in the milk of Australian monotremes such as the short-beaked echidna [69]. We solved the crystal structure of the HAdV-D36 fiber knob in complex with 4-*O*-Ac-3'SL isolated from echidna milk, and we observed that the lactose component is not ordered in two of the three complexed glycans and does not seem to contribute to the binding. The only detectable contact mediated by the glycan stem is formed between the poorly ordered *O*6 atom of the galactose moiety and a backbone carbonyl function (**Fig 5A** and **5B**). Thus, the Neu4,5Ac$_2$ part must be the decisive factor for the increased binding. As 4-*O*-Ac-3'SL is the only Neu4,5Ac$_2$-containing glycan in the array, we conclude that this SA variant is probably recognized in several glycan contexts, similar to what has been found for HAdV-D37, which recognizes a range of Neu5Ac-containing glycans. For the subsequent studies, we, therefore, replaced 4-*O*-Ac-3'SL with the more readily accessible synthetic α-2-*O*-methyl-Neu4,5Ac$_2$. The structures of SA and α-2-*O*-methyl-Neu4,5Ac$_2$ in complex with HAdV-D36 FK are relatively similar (**Fig 5C**). However, due to the narrow binding pocket of the HAdV-D36 FK, the three neighbouring Neu4,5Ac$_2$ moieties seem to form a

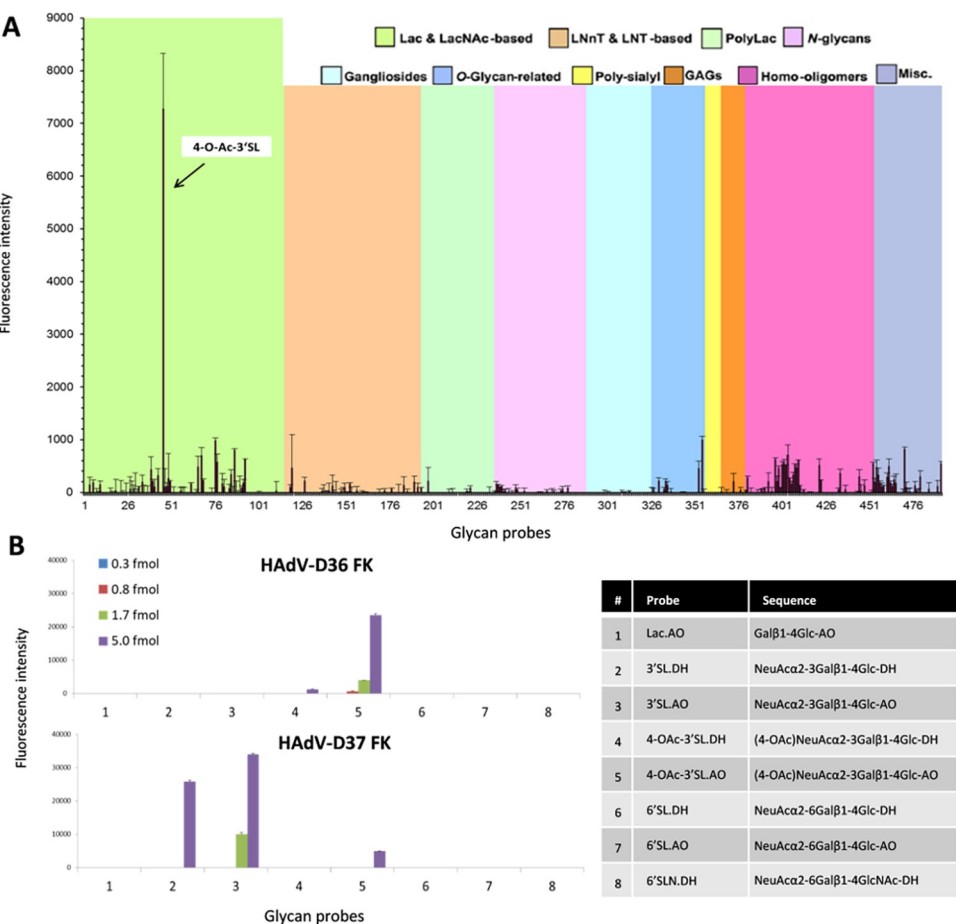

**Fig 4. Glycan array screening of the HAdV-D36 FK and comparison with the HAdV-D37 FK.** (*A*) A broad-spectrum glycan screening array containing 492 glycans (see legend on the upper right and **S1 Table**) showed a low overall signal with one outstanding signal elicited by 4-*O*-Ac-3'SL (7,279 fluorescence units). Numerical scores for the binding intensity are shown as means of fluorescence intensities of duplicate spots at 5 fmol/spot, with error bars representing half of the difference between the two values. (*B*) A small, focused array designed to test for HAdV-D36 FK binding to 3'SL variants. Lipid-linked glycan probes with their corresponding sequences, including lipid tags are listed on the right panel (more information on lipid tag in **S4 Table**). The doses of the glycan probes arrayed per spot are indicated. Notably, HAdV-D36 FK demonstrated selective binding exclusively to 4-*O*-Ac-3'SL, contrasting with the preference of HAdV-D37 FK for 3'SL. Linker (or tag) effects in glycan binding are commonly observed in glycan array studies and are likely associated with glycan presentation [117]. Error bars represent half of the difference between the two values.

triangular hydrophobic contact through the 4-*O*-acetyl groups (green dashes) that is not possible for Neu5Ac (**Fig 5D**). The crystal structure of the HAdV-D36 FK in complex with another synthetic SA variant, α-2-*O*-methyl-Neu4,5,9Ac$_3$, shows that additional acetylation at position 9 can be accommodated, but does not contribute additional interactions to the binding (**S9A Fig**). However, none of the three Neu5,9Ac$_2$-containing probes showed binding to HAdV-D36 fiber knob in the glycan microarray (**S2 Table**). Molecular modeling of SA variants containing an additional *O*-acetylation at positions 7 and 8 suggests that both would likely induce clashes and would result in an abrogation of the binding (**S9B Fig**). We, therefore, conclude that 4-*O*-acetylation has a specific effect on HAdV-D36 recognition that is not seen with other SA variants.

## Structural basis for the Neu4,5Ac$_2$ preference

HAdV-D36 FK and Neu4,5Ac$_2$ binding mode is similar to that of Neu5Ac. The key polar interactions mediated by the carboxyl and amide functions of Neu5Ac are also found in the Neu4,5Ac$_2$ complex structure, as are the residues that make up the hydrophobic cavity for the N-acetyl group and the water-bridged hydrogen bond mediated by the glycerol function. The binding environment of the *O*-acetyl groups is made up of three amino acids: N313, Y315, and P323). Y315 thereby acts as a 'gatekeeper' residue that inserts itself between the *N*-acetyl and *O*-acetyl groups of Neu4,5Ac$_2$ (Fig 5E). Even a subtle displacement of this residue would likely interfere with Neu4,5Ac$_2$ binding, as shown in other cases [70,71]. N313 is one of the residues of the conserved *VSN* stretch and closes the binding pocket from below. Similarly to the Neu5Ac complex structure, N313 engages in a direct hydrogen bond with the β-face of one Neu4,5Ac$_2$ moiety (S10A Fig). The three N313 residues of the trimeric knob are located right at the threefold axis, and each residue contacts two Neu4,5Ac$_2$ residues simultaneously via van der Waals interaction (S10B Fig). The topology and reduced polarity of this residue allow the formation of the triangular inter-sugar hydrophobic contact. Interestingly, the *O*-acetyl function does not contribute any polar contact and instead induces the tilting movement under the steric influence of the bulky side chains of Y315 and P323. This interaction is supposedly not favorable for the binding; however, the steric constraints bring the *O*-acetyl groups of the three

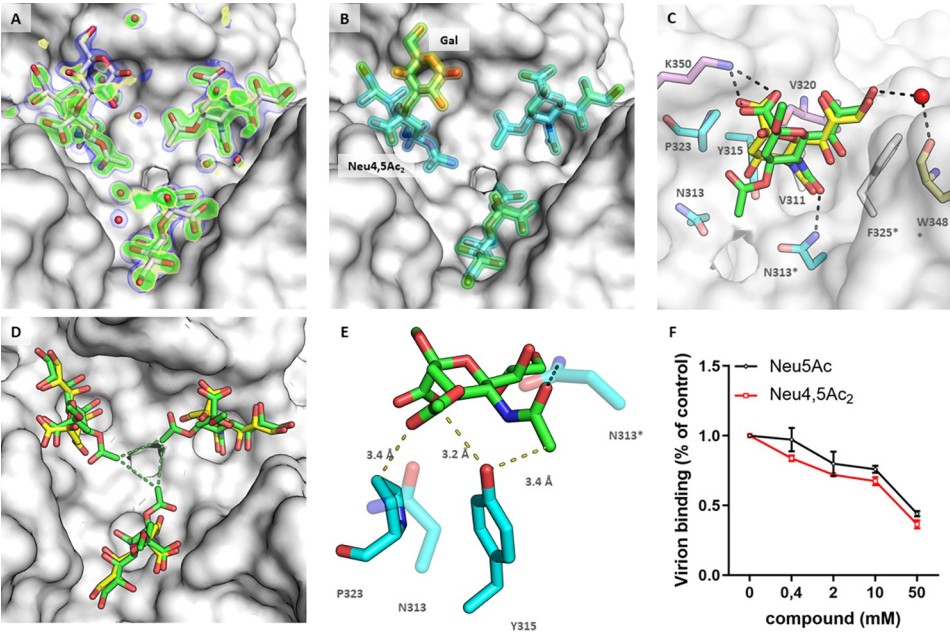

**Fig 5. Structure of the HAdV-D36 FK in complex with 4-*O*-Ac-3'SL and comparison of Neu5Ac and Neu4,5Ac$_2$ engagement as well as functional relevance.** (*A*) Simulated annealing omit map (green: 3σ; yellow: 2.5σ) and 2F$_o$-F$_c$ map after refinement (1σ, blue) of 4-*O*-Ac-3'SL in complex with the HAdV-D36 FK. The lactose stem is generally poorly ordered and only partially visible in one of three cases. (*B*) 4-*O*-Ac-3'SL colored by -factors ranging from 10Å$^2$ (blue) to 60 Å$^2$ (red). (*C*) Superposition of the Neu5Ac (yellow) and Neu4,5Ac$_2$ (green) complex structures. Neu4,5Ac$_2$ is forced towards the threefold axis by the steric influence of N313, Y315, and P323, while the polar key contacts are only slightly altered. (*D*) This movement causes the three neighboring Neu4,5Ac$_2$ moieties to form a triangular hydrophobic contact (green dashes) that is not possible for Neu5Ac. (*E*) Steric influence of Y315 and P320 on Neu4,5Ac$_2$. Short-ranged van-der-Waals contacts are formed with both acetyl groups of the sugar. As a result, even slight alterations in the Y315 conformation would interfere with the binding. (F) Binding of preincubated HAdV-D36 virions with Neu5NAc and Neu4,5NAc2 to A549 cells. Data are presented in % of control, here control refers to the binding of virions to cells without Neu5Ac or Neu4,5NAc2. The data represent values from one experiment with triple technical replicates. Error bars represent mean ± SD.

sugar moieties into a position in which they contact each other by means of a triangular hydrophobic interaction (mean distance between methyl C atoms: 4.3 Å) that displaces water molecules in the area (**S10B** and **S10C Fig**). A similar effect is not expected for Neu5Ac. Interestingly, all known complex structures with 4,5-acetylated SA display a similar arrangement (**S11A–S11D Fig**). In the complex structure of HAdV-D37 [47], the steric influence of the two residues Y312 and P320 is observable to the same extent, but the overall arrangement of the knob trimer does not allow the formation of this hydrophobic interaction (mean distance between methyl C atoms: 5.2 Å) (**S10D** and **S10E Fig**). This finding is in line with the preference for Neu5Ac over Neu4,5Ac$_2$ observed for HAdV-D37. As a result, the observed relative twist between Neu5Ac and Neu4,5Ac$_2$ is much less pronounced for HAdV-D37 than for HAdV-D36 (**S10C** and **S10F Fig**). It is tempting to speculate that the presence of this direct interaction between the three sugars might lead to a cooperative effect that alters the binding kinetics and accounts for the observed binding preference of HAdV-D36, especially in the absence of the strong electrostatic attraction between knob and carbohydrate observed for HAdV-D37.

To provide additional support for our hypothesis that the relative arrangement of the monomers is the main cause of the difference in Neu4,5Ac$_2$ binding, we replaced the central three amino acid stretch of HAdV-D36 (VSN) with that of HAdV-D37 (YGT) and *vice versa*. Indeed, the mutations assimilated the spatial arrangement at the central cavity (**S12A and S12B Fig**). As a result, the HAdV-D37 *VSN* mutant restored the triangular hydrophobic contact (**S13 Fig**, mean distance between methyl C atoms: 4.4 Å), although the electron density for the sugar was not as pronounced as for the HAdV-D36 wild type. The HAdV-D36 FK *YGT* mutant, on the other hand, displayed a widened binding pocket as compared to the wild type. The mutant knob still engages Neu5Ac, but completely lost its ability to bind Neu4,5Ac$_2$ due to the loss of the water-bridged contact with T313*, which distorts the sugar orientation and produces clashes with the gatekeeper residue Y315 as well as P323 and T313 (**S12C–S12G Fig**). Although the *VSN* trimer-interface configuration is highly conserved among HAdVs of the 'HAdV-D36-like' clade, they are not under general suspicion of engaging Neu4,5Ac$_2$. As an example, the low seroprevalence type HAdV-D48 [72] exhibits an altered spatial arrangement of the lysine at the position that usually mediates the key contact to the sugar's carboxyl group (**S6B Fig**) [57]. This displacement, caused by a minor rearrangement of the IJ loop and a mutation at position 327, results in a distorted binding site and a complete abrogation of Neu5Ac binding. The HAdV-D26 FK belonging to the 'HAdV-D37'-like clade, in turn, exhibits a less displaced lysine residue at the respective position and maintains its ability to bind Neu5Ac (**S6C Fig**) [49]. However, this movement results in a somewhat different binding orientation that would require a conformational flexibility of the gatekeeper tyrosine and surrounding residues in a Neu4,5Ac$_2$ context to prevent clashes. These examples demonstrate the level of sophistication needed to maintain the three-dimensional arrangement of HAdV's SA-binding sites that are sensitive to subtle alterations affecting the quaternary structure or secondary mutations that influence the binding site topology without directly interfering with the binding. In this light, it is of little surprise that HAdV-D36 is the only HAdV type known to efficiently engage Neu4,5Ac$_2$ to date. To validate the structural data showing preferential binding to Neu4,5Ac$_2$, a functional binding assay was performed (**Fig 5F**). Knowing that HAdV-D36 is the only HAdV that binds efficiently to Neu4,5Ac$_2$, we analysed the binding of HAdV-D36 pre-treated with Neu5Ac and Neu4,5Ac$_2$ to A549 cells. In contrast to the experiment performed in **Fig 2B**, we used Alexa-488 labelled HAdV-D36 viruses in place of $^{35}$S-labeled. Both monosaccharides inhibited HAdV-D36 binding, and with similar efficiency. Neu4,5Ac$_2$ pre-treatment was slightly more efficient in inhibiting virion attachment, although the difference is not statistically significant. However, it should be noted that glycans on the

cell surface are typically branched presented in a multivalent manner, either as glycolipids or at multiple sites on proteins such as mucins, thus allowing for an avidity effect. This aspect was not evaluated in this experiment, and it can greatly enhance the subtle differences in binding and inhibition.

## Discussion

The systematic analysis of possible HAdV-D36 attachment factors revealed that virion attachment to various cell types depends on CAR and SA-containing glycoproteins. The attachment is sensitive to the treatment of cells with proteases and neuraminidase. It is furthermore diminished by inhibitors of *O*-glycosylation and, to a lesser extent, *N*-glycosylation, and can be competed with soluble Neu5Ac analogues. However, we were never able to inhibit the binding by more than 80% (**Fig 2D**). Thus, although CAR and Neu5Ac-containing glycoproteins appear to be the major factors for HAdV-D36 cell attachment, there are likely additional, yet uncharacterized factors that can account for about 20% of the binding, at least on A549 cells. This finding is not unprecedented, as HAdV-D49, which possesses a similar FK, is reported to use a yet unknown cell surface receptor for entry [73]. In the light of the results presented here, an unambiguous preference for CAR and SA shown by fiber knobs (**Fig 1C**) indicates that the additional attachment factor is most likely recognized by a different capsid protein. Recently, the hexon of HAdV-D56 was shown to directly bind to CD46 for cell attachment and entry [34]. However, in this case CD46 was shown to be the major attachment factor, while both CAR and SA did not mediate infection. Hence the overall attachment mechanism is at least somewhat if not completely different. Species D HAdVs, can also engage $\alpha_v$ integrins by their penton base, for cell attachment [74]. The penton base sequence of HAdV-D36 is similar to that of HAdV-D9, and a similar mechanism seems possible for A549 cells. Therefore, we suggest that there are at least four different attachment factors used by species D HAdVs. Coagulation factor X, which is known to engage the HAdV hexon and mediate cell uptake, does not bind the HAdV-D36 particle (**S2H Fig**).

Our cell-based assays show that HAdV-D36 recognizes CAR with its FK domain, and our structural data indicate that it probably possesses an altered binding interface compared to established CAR-binding knobs such as HAdV-C5 FK. Although we could clearly demonstrate that CAR is functional in supporting HAdV-D36 attachment to different cell types, whether CAR can serve to mediate cell infection, as well, remains to be clarified. In the case of the closely related HAdV-D37, which recognizes CAR with its knob domain and can use it for cell attachment, but not for infection, the stiffness of the fiber shaft does not allow an efficient interplay between CAR and integrins on the cell surface [75]. As discussed above, HAdV-D36 possesses a similar shaft sequence and misses known binding motifs, as well. Therefore, CAR recognition of both viruses might be more important for virus spread after lytic completion of the life cycle than for the actual infection process [76]. This notion agrees well with the finding that both HAdV-D36 and–D37 FKs additionally attach to glycoproteins, which are also much more prominently displayed on the apical side of epithelial layers than CAR.

Glycan array screening and structural investigations indicate that the rare Neu5Ac variant Neu4,5Ac$_2$ is engaged more effectively by the knob of HAdV-D36 than the more common Neu5Ac (**Figs 4, 5, and S10**). We have identified the factors responsible for this preference and identified residues that influence the binding by altering the knob topology. Analysis of the knob phylogeny suggests that these residues are evolutionarily conserved in about half of species D, although Neu4,5Ac$_2$ binding is likely a specific ability acquired only by HAdV-D36 or a small subset of HAdVs of the 'HAdV-D36-like clade'. The specific exploitation of SA variants by human or animal viruses is not unprecedented, and the presence or absence of these

derivatives in different organisms determines the tropism of many SA-binding viruses (reviewed in [77]). A prominent example is human influenza C virus, which uses $Neu5,9Ac_2$ as a receptor and possesses hemagglutinin with an esterase activity as receptor-destroying enzyme [78]. Some toro- and lineage A betacoronaviruses have long been known to specifically use *O*-acetylated Neu5Ac variants as receptors on the cell surface *via* their hemagglutinin-esterase (HE) proteins (reviewed in [79]). Although $Neu5,9Ac_2$ is the prototypical receptor for coronaviruses, some virus strains have developed altered specificities. For example, two biotypes of mouse coronaviruses are distinguished. While members of group I use $Neu5,9Ac_2$, some viruses constituting group II evolved to specifically use $Neu4,5Ac_2$. Mouse hepatitis virus strain S (MHV-S), a murine coronavirus of class II, has been demonstrated to establish this receptor switch by modest changes in both the lectin and esterase domain that allow the formation of a binding pocket optimal for the spacing of its two acetyl groups [70,71]. This arrangement resembles the pocket formed by the protein and two neighboring residues in the HAdV-D36 complex structure. For these viruses, the presence of $Neu4,5Ac_2$ in the colon and brain is likely a determinant of tropism [80]. Since $Neu4,5Ac_2$ is abundant on fish skin [81], several piscine viruses including infectious salmon anemia virus (ISAV) use this receptor as well [82]. $Neu4,5Ac_2$ from the horse and guinea pig sera influence the tropism of some influenza A H2 and H3 strains by inhibiting their neuraminidase [83]. In fact, this mode of action is assumed to be one of the physiological functions of $Neu4,5Ac_2$ [77,84]. A potential physiological role of $Neu4,5Ac_2$ for human infection is currently unclear, however the dietary intake of $Neu4,5Ac_2$ can be one possible way for its presence in human tissues similar to Neu5Gc [85]. $Neu4,5Ac_2$ has not yet been detected in human tissue samples, its occurrence and distribution in humans are still not understood. This form of SA is difficult to analyze histochemically due to the acid/base liability of the 4-*O*-ester group (reviewed in [77,84]). Suitable detection methods have been developed only relatively recently, and tissue staining using viral HE proteins has so far not covered all tissues [81,86]. Furthermore, studies in mice suggest that its synthesis is restricted to only a few tissue types, such as . . . [86,87]. In this light, it is possible that $Neu4,5Ac_2$ might be expressed in specialized tissues that have not yet been investigated in this regard. We tested the functional relevance of 4-*O*-acetylation of Neu5Ac (using 2-*O*-Me analogue of $Neu4,5Ac_2$) in HAdV-D36 attachment to A549 cells. We observed a small but not statistically significant increase in the degree of inhibition for HAdV-D36 binding to cells when virions were pre-treated with $Neu4,5Ac_2$ monosaccharides compared to virions pre-treated with Neu5Ac. However, the discrepancy might arguably be enhanced *in vivo* by the avidity effect imparted by the clustered presentation of complex SA capping glycans on the cell surface. Along these lines, our glycan arrays, containing lipid-linked glycan probes presented in an oriented and multivalent state with potential mobility on nitrocellulose matrices, showed a significantly more pronounced difference in signal.

From a pathogen's perspective, specifying on SA variants with a restricted expression profile may help to circumnavigate the vast amounts of non-productive 'decoy' sialylated glycotopes that are present on virtually any tissue in high local concentrations. Since HAdVs do not possess a receptor-destroying enzyme, it appears as a feasible strategy for HAdV-D36 to bind effectively to a spatially confined epitope while having a lower affinity for the more ubiquitous variant. The preference of $Neu4,5Ac_2$ by HAdV-D36 and the distribution of the sugar among different species overlap remarkably well with the animal range that is reported to be permissive for this virus. It is therefore tempting to speculate that HAdV-D36 might be the first HAdV known to possess an animal reservoir. Following this rationale, it also appears plausible that HAdV-D36 has maintained its ability to engage Neu5Ac and additionally binds a protein receptor. It should not go unnoticed that the presence of an animal host reservoir would directly imply a route of virus transmission through contact with domestic animals or the

consumption of livestock, respectively. Many human viruses are known to infect intermediary animal hosts, and adenoviruses are known to be spread through foodborne oral-fecal transmission [88]. In particular, chickens seem to be a suitable candidate for such a reservoir due to the practice of intensive mass animal farming and the fact that they are one of the main sources of meat in many countries. However, there is no publicly available data for HAdV-D36 seroprevalence in animals to date. Such data would give insights into the physiological relevance of Neu4,5Ac$_2$ for human HAdV-D36 infection.

## Materials and methods

### Cells, and viruses

Human lung carcinoma cells A549 (a gift from Dr. Alistair Kidd) were grown in Dulbecco's modified Eagle medium (DMEM; Gibco, Paisley, UK) supplemented with 10% fetal bovine serum (FBS; Thermo Scientific, Cramlington, UK), 20 mM HEPES (Fischer Scientific, Fair Lawn, USA), and 20 U/mL penicillin + 20 μg/mL streptomycin (GE Healthcare, South Logan, USA). Chinese hamster ovary cell CHO-K1 (gift from David Fitzgerald) were grown in Ham's F12 medium (Gibco) supplemented with 10% FBS (Thermo Scientific) and 20 U/mL penicillin + 20 μg/mL streptomycin (GE Healthcare). CHO-MOCK, CHO-CAR, CHO-CD46, CHO-LEC2, CHO-Pro-5, CHO-CD21 474, CHO-CD21 491, FHS, EKVX, HEP-G2, HEK 293, 3T3-L1, and SGBS s were grown as described in the references listed [29,89–97]. HAdV-D36 (strain 275) was mainly used in all the experiments. In some experiments, HAdV-C5 (strain Adenoid 75, source ATCC) and HAdV-D37 (Strain 1477) were used as compound controls. These viruses were produced in A549 cells with or without [35]S labeling and purified on cesium chloride (CsCl) gradients as described previously [98]. For one experiment, HAdV-D36 viruses were labelled with Alexa Fluor 488 (AF488) as described in [99]. The detailed protocol for virus production and labelling is provided in S1 Text.

### Cell-binding assays

Binding assays using [35]S-radiolabeled HAdVs and AF-488 HAdV-D36 were carried out as described in detail in S1 Text. Briefly, the cells were detached with PBS containing 0.05% EDTA and reactivated for 1 h at 37°C in growth media. Cells (1x10$^5$ cells/well) were added to a V-shaped bottom 96-well plate and after washing once with binding buffer (BB: DMEM (Gibco) supplemented with 20 mM HEPES (Fischer Scientific), 20 U/mL penicillin + 20 μg/mL streptomycin (GE Healthcare) and 1% bovine serum albumin (Roche, Mannheim, Germany)), incubated with virions (HAdV-D36: 1x10$^9$ virions/well, HAdV-C5: 1x10$^9$ virions/well and HAdV-D37: 5x10$^8$ virions/well) for 60 min on ice. Non-bound virions were then removed by washing with binding buffer. Samples were analyzed by measuring radioactivity in a scintillation counter (1450 Microbeta, Wallac) and fluorescence by flow cytometry.

### Hemagglutination assay

Animal red blood cells were kindly provided by Agrisera (Vännäs, Sweden). Bovine, chicken, equine, goat, porcine and rabbit blood were collected in sodium citrate. Human venous blood was collected in sodium citrate from three volunteer donors with different blood types according to the AB0 system (A, B, and O). Washed erythrocytes from blood samples were diluted to a 1% (v/v) solution in PBS cells were counted. Fifty microliters PBS were added to a round bottom 96-well plate. Virions (in a range of 1x10$^9$ to 3.2x10$^{11}$ depending on virus identity) followed by 2-fold serial dilution were incubated with fifty microliters of the 1% cell suspension for 1 hour at room temperature. The results were interpreted visually as complete

agglutination when a lattice of red blood cells, hemagglutination negative when buttons are formed in the bottom of the wells. If buttons were not running when the plate was tilted, they were judged as incomplete agglutination, running buttons were considered definitely negative. To estimate the strength of the interaction between virions and cells, the number of virus particles per cell needed to cause hemagglutination, which is one hemagglutinating unit (vp/HAU) was calculated.

## Cloning, purification, and crystallization of HAdV fiber knobs and their complex structures

The construct design and cloning of HAdV-D36 FKs was analogous to that of the HAdV-D37 FK as described previously [100]. The HAdV-D36 FK was amplified from HAdV-D36 strain 275 genomic DNA using the following primers: 5'-gcc cat ggg aga ctt agt agc ttg-3' (forward); 5'-cgc ctc gag tca ttc ttg agc gat ata tga gaa ag-3' (reverse) and cloned into a pPROEX htb vector. The YGT and VSN mutants were prepared by site-directed mutagenesis using a modified Strategene protocol (see details in S1 Text). All construct sequences were verified by Sanger-sequencing (performed at MWG Eurofins) and comparison to GenBank entries GQ384080.1 (HAdV-D36 complete genome) and ABK59080.1 (HAdV-D37 fiber). Expression and purification of the HAdV-D36 FKs were performed essentially as reported previously for HAdV-D37 [100]. Cloning, expression and purification of the HAdV-B11 FKs were performed as described previously [56].

For crystallization, the HAdV-D37 FK was concentrated to 12–13 mg/mL, and the HAdV-D36 FK to 8.0–8.2 mg/mL. The HAdV-D37 FK was crystallized as described previously [100]. The HAdV-D36 FK was crystallized by the hanging drop vapor diffusion method using 23–26% PEG 3,350, 175–200 mNH4Ac, 0.1 M Bis-Tris pH 5.5 and an initial drop size of 1+1μL at 4˚C. Complex crystals were prepared by co-crystallization with 10 mM of the respective carbohydrate compound. Crystals were flash frozen in liquid nitrogen without cryoprotection. Data collection was carried out at the beamlines X06SA and X06DA at the SLS (Villigen, Switzerland). Data were processed with XDS [101,102]. For the initial structure of HAdV-D36 wt, the phase problem was solved using Molrep and a HAdV-D37 FK-based model prepared with CHAINSAW [103]. If possible, subsequent structures were solved by simple rigid body refinement and simulated annealing in phenix.refine [104]. Otherwise, the phase problem was solved with Molrep [105] or Phaser [106]. Refinement was carried out using Coot [107] for real space refinement and phenix.refine with automatically determined non-crystallographic symmetry (NCS) restraints and isotropic B-Factor refinement. At high enough resolution, translation-libration-screw (TLS) tensors (all structures in S3 Table) or anisotropic B-Factors were refined. Ligands were unambiguously placed into Fo-Fc difference maps and refined using restraints from the CCP4 monomer library. Waters were located using an automated algorithm in Coot. Simulated annealing omit maps were calculated with phenix.refine and FFT [108].

## Protein structure accession numbers

Coordinates and structure factor amplitudes have been submitted for deposition in the Protein Data Bank (PDB) under accession numbers 9FAE (HAdV-D36 in complex with 4-$O$-acetyl-3'-sialyllactose), 9FAF (HAdV-D36 in complex with Neu4,5Ac$_2$), 9FAG (HAdV-D36 in complex with Neu4,5,9Ac$_3$), and 9FAH (HAdV-D37 in complex with Neu4,5Ac$_2$).

## Superpositioning of complex structures

Superpositions of FKs and the corresponding complex structures were performed using the 'align' algorithm in PyMol (The PyMOL Molecular Graphics System, Version 1.8 Schrödinger,

LLC). For HAdV-D26, HAdV-D37, and HAdV-D48 the PDB entries 6qu8, 1uxa, and 6fjq, respectively, were used for structural coordinates. An important distinction has to be made between aligning all atoms in the knob trimer and alignment based on single chains.

## Complexation of HAdV-D36 FK with CAR-D1 and CD46-D4

The complexation experiments were carried out as described previously [55] with minor alterations. Purified HAdV-D36 FK was mixed with CAR-D1 (laboratory stock produced by A. Thor) and CD46-D4 (purified as described previously [109]) in different molar ratios in their respective buffers and incubated for 20 min at 4˚C. 30 µL were subjected to analytical size exclusion chromatography using a Superdex 200 column on an ETTAN system (GE Healthcare, Sweden) using a standard buffer containing 30 mM Tris-HCl (pH 7.5) and 150 mM NaCl. The peaks were analyzed by comparison to a standard laboratory calibration curve and, if possible, by SDS-PAGE. In the case of CAR-D1, both HAdV-D36 FK and CAR-D1 alone were run at the same concentrations for comparison. In the case of CD46-D4, a HAdV-B11 FK [56] complexed with CD46-D4 was run as a positive control instead of CD46-D4 alone.

## Nano differential scanning fluorimetry (nanoDSF)

Samples were diluted to a concentration of 0.1 mg/mL in a buffer containing 150 mM NaCl, 20 mM imidazole, and 30 mM Tris-HCl, pH 7.5. The experiments were performed on a NanoTemper Prometheus using a sample volume of 10 µL per capillary. The capillaries were filled directly from the respective solution. An initial fluorescence scan of loaded capillaries was performed at 20˚C, low gain sensitivity in order to ensure that samples were within the optimal concentration range. The nanoDSF experiment was performed in a single run by heating all samples from 20˚C to 95˚C with 1˚C/min. To determine the melting point (Tm), the shift in native tryptophan fluorescence was monitored by plotting changes in the emission at 350 and 330 nm.

## Glycan array screening

His-tagged HAdV-D36 FK was analyzed in two different array sets: the broad-spectrum screening array set containing 492 sequence-defined lipid-linked glycan probes (in-house designation 'Sequence-defined Array Sets 32–39') and a focused sialyl glycan array set containing 8 oligosaccharide probes in dose-response format (see **Fig 4B**). The probes were robotically printed in duplicate on nitrocellulose-coated glass slides at the levels indicated using a non-contact instrument [110]. Details of the preparation of the glycan probes and microarrays are available in **S4 Table** in accordance with the MIRAGE guidelines (Minimum Information Required for A Glycomics Experiment)[111]. The microarray analyses were performed essentially as described previously [112]. In brief, microarrays were blocked in 0.3% (v/v) Blocker Casein (Pierce), 0.3% (w/v) bovine serum albumin (Sigma A8577) in HEPES buffered saline (5 mM HEPES, pH 7.4, 150 mM NaCl, 5 mM CaCl2). The His-FK proteins of HAdV-D36 and HAdV-D37 were tested as protein-antibody complexes that were prepared by pre-incubating FK protein with mouse monoclonal anti-polyhistidine and biotinylated anti-mouse IgG antibodies (both from Sigma) at a ratio of 4:2:1 (by weight) and diluted in blocking solution to provide a final FK concentration of 150 µg/mL. Binding was detected using AF647-labeled streptavidin from Molecular Probes (1 µg/mL). In the small sialyl oligosaccharide array, His-tagged HAdV-D36 FK and HAdV-D37 were also tested under the same condition. The oligosaccharide probes are all lipid-linked, neoglycolipids (NGLs) or glycosylceramides, and are from the collection assembled in the course of research in the Imperial College London Glycosciences Laboratory. For the definition of the lipid moieties of the probes, please see https://

glycosciences.med.ic.ac.uk/docs/lipids.pdf. Imaging and data analysis are described in the Supplementary MIRAGE document (**S4 Table**). Numerical scores for the binding intensity are shown as means of fluorescence intensities of duplicate spots, with error bars representing half of the difference between the two values.

## Purification of 4-O-acetyl-3'-sialyllactose (4-O-Ac-3'SL) from echidna milk oligosaccharides (EMOs)

4-O-Ac-3'SL was isolated from an oligosaccharide mixture derived from the milk of the Australian short-beaked echidna. EMOs [113] were fractionated by gel filtration chromatography on a Bio-Gel P4 column (16 x 90 cm) with elution by ammonium acetate (0.2 M) and detection by refractive index (**S14A Fig**). See detailed description in S1 Text. Quantitation was carried out by microscale orcinol assay as essentially as described [114].

## Synthesis of differentially O-acetylated α-2-O-methyl-Neu5Ac variants

Methyl 5-acetamido-4-$O$-acetyl-3,5-dideoxy-D-glycero-a-D-galacto-2-nonulopyranosidonic acid (2-$O$-Me-Neu4,5Ac$_2$) and methyl-5-acetamido-4,9-di-$O$-acetyl-3,5-dideoxy-D-glycero-a-D-galacto-2-onulopyranosidonic acid (2-$O$-Me-Neu4,5,9Ac$_3$) were synthesized essentially as described previously [115].

### Statistical analyses

The results are shown as mean, and the error bar represents SD of mean biological or technical replicates as indicated in the figure legend. The statistical analyses of significance between two groups was determined by unpaired two-tailed t test. $P < 0.05$ was considered to be significant. The p values represented as *$p < 0.05$, **$p < 0.01$, ***$p < 0.001$, ****$p < 0.0001$; ns represents nonsignificant. We performed the statistical analyses by using GraphPad Prism 8.

### Supporting information

**S1 Fig. HAdV-D36 binds better to A549 cells than to other cell lines. A** Binding of $^{35}$S-labelled HAdV-D36 virions to different human-derived cell lines A549 (adenocarcinomic human alveolar basal epithelial cells), EKVX (human lung adenocarcinoma cell line), HEK293 (human embryonic kidney 293 cells), FHS (human small intestine cell line), HEP-G2 (human hepatocellular carcinoma cell line). **B** Binding of $^{35}$S-labelled HAdV-D36 virions to A549, 3T3-L1 (mouse fibroblasts that can differentiate into an adipocyte-like phenotype) and SGBS (human Simpson-Golabi-Behmel syndrome preadipocyte cell line) cells. The amount of virus particles bound to cells was presented as counts per minute (CPM). All experiments were performed three times with duplicate samples. Error bars represent mean ± SD. The statistical significance was determined by unpaired two-tailed t test; * P of $< 0.05$.
(TIF)

**S2 Fig. Evaluation of HAdV-D36 binding to A549 cells.** Binding of $^{35}$S-labelled HAdV-D36 virions to A549 cells treated with different proteases. **A** Proteinase K (a broad-spectrum serine-protease), **B** Bromelain (a mixture of pineapple cysteine proteases), **C** V8 (an endo-proteinase Glu-C serine protease cleaving after glutamic acid), **D** tunicamycin (an inhibitor of N-glycosylation), **E**, P4 (DL-threo-1-phenyl-2-palmitoylamino-3-pyrrolidino-1-propanol, an inhibitor of glycolipid synthesis) and **F** heparinase III (10 mU/μL, which cleaves heparan sulfate from cell surface). **G** Attachment inhibition assay with soluble heparin (M$_w$ approx. 21000). HAdV-D37 was used as a positive control. **H** $^{35}$S-labeled HAdV-D36 and HAdV-C5 virions, pre-incubated with physiological concentrations of coagulation factor X (FX: 10 μg/

mL), binding to A549 cells. In figure A-F and H, y-axis shows the amount of virus particles bound to cells and represented as CPM (count per minute). All experiments were performed three times with duplicate samples. Error bars are representing mean ± SD. The statistical significance was determined by unpaired two-tailed t test; ns = not significant, * P of < 0.05, ** P of < 0.01 and *** P of < 0.001.
(TIF)

**S3 Fig. Differential Scanning Fluorimetry of the HAdV-D36FK.** The scan was performed using a Prometheus NT.48. The ratio of the fluorescence at 350 nm and 330 nm is plotted against the temperature (upper part). The first derivative calculation can be deduced to determine the Tm (lower part). At pH 7.5 (red and green curves, n = 5), a Tm of 86.3–86.5˚C was observed. At a pH of 2–3 (blue), the FK stability was lowered, but still high (Tm = 62.4˚C). The experiment was performed in the course of a Prometheus demonstration by Dr. Fabian Zehender (NanoTemper Technologies GmbH).
(TIF)

**S4 Fig. Molecular modeling and SEC analysis of CAR and CD46 binding. A Superposition of** HAdV-D36 FK (red) onto the HAdV-A12 FK (blue) in the CAR (light gray) complex structure (PDB-ID 1KAC). The HAdV-D36 FK heavily clashes with CAR through its DG loop (red dotted line). **B** Superposition of HAdV-D36 FK (red) onto HAdV-B11 (green) in the CD46 complex structure (PDB-ID 2O39). The HAdV-D36 DG loop is located proximally in front of CD46 (dark gray) and would likely interfere with CD46 binding without producing direct clashes. Superpositions of fiber knobs and the corresponding complex structures were performed using the 'align' algorithm in PyMol. C Conservation of the CAR-interface between HADV-D36 and HAdV-D37. The surface of HAdV-D37 FK (PDB 2J12) is shown in surface representation, and the CAR interface is colored blue. The residues of HAdV-D37 that form the interface are shown in teal. HAdV-D36 FK was superposed using the monomers, and its residues are shown in white. Residues that are only functionally conserved are colored yellow, residues that are not conserved are colored orange. D The canonical interface between HAdV-D36 FK and CAR is disturbed by the DG loop. The structure of HAdV-D36 was superposed on the HAdV-D37 FK CAR complex (PDB 2J12), superposing the monomers. HAdV-D36 is shown in white, CAR in gold. Although the resulting interface is highly shape complementary, the HAdV-D36 DG loop from the adjacent chain (red) is disturbing the interaction. **E** SEC of the HAdV-D36 FK / CAR complex (red line). HAdV-D36 FK (blue line) was incubated with soluble CAR-D1 (green line) for 20 minutes prior to SEC. The peak fractions (orange bar) of SEC were characterized by SDS-PAGE shown in inset. Equal amounts of FK and CAR were used for optimal comparability. **F** Comparison of SEC profile of complex formation of HAdV-D36 (blue line) with CD46 (green line) to HAdV-B11 with CD46. The elution profile of HAdV-D36 FK-CD46 is shown with red line while the green line depicts that of HAdV-D36 FK-CD46. The proteins causing the respective peaks are noted below the chromatograms.
(TIF)

**S5 Fig. Comparison of DG loop and trimer interface conservation, and Neu5Ac binding among HAdV species D. A** Excerpt of the alignment of all species D HAdVs from type HAdV-D08 to D56 performed with Clustal Omega [116]. The 18 HAdV types belonging to the 'HAdV-D36-like' clade possess elongated DG loops (yellow) and a VSN or VTN interface configuration (green). The 17 types belonging to the 'HAdV-D37-like' clade have shorter DG loops (orange) and a YGT (light pink) or YGN (purple) interface configuration. **B** The VSN trimer interface (green) as found in the HAdV-D36 FK. **C** YGT trimer interfaces (light pink)

as observed in the HAdV-D37 FK (PDB-ID 1UXE). **D** Neu5Ac (yellow) binding mode observed for the HAdV-D36 FK (gray surface). The structure was solved with α-2-*O*-methyl-Neu5Ac, and methyl functions have been left out for clarity. Residues marked with an asterisk are contributed by the clockwise neighboring monomer. Key contacts are mediated by K350 and V320 (purple), while the hydrophobic binding cavity for the *N*-acetyl function is formed by V311, V327*, and F352*. W348 (yellow) contacts the glycerol function by a water-bridged contact. The three residues N313, Y315, and P323 (cyan) put steric constraints onto the sugar's O4 atom, while N313* contacts the *N*-acetyl group by means of a hydrogen bond from the α-face. The α- and β- faces of Neu5Ac are marked with an arrow. **E** Neu5Ac binding mode observed for HAdV-D37 (PDB-ID 1UXA). Coloring according to A. Unlike N313 in HAdV-D36, T310 (yellow) does not put steric pressure onto the Neu5Ac O4, and T310* contacts the sugar via a water-bridged polar contact. **F** Relative placement of Neu5Ac complexed to HAdV-D36 FK (yellow) and HAdV-D37 FK (teal) upon superposition of the knob trimers. The sugars show a prominent relative shift.
(TIF)

**S6 Fig. Conservation of SA binding among species D HAdVs. A** Comparison of SA binding of HAdV-D36 with HAdV-D37 upon superposition of only one monomer. B Comparison of SA binding of HAdV-D36 with HAdV-D48 upon superposition of only one monomer. C Comparison of SA binding of HAdV-D36 with HAdV-D26 upon superposition of only one monomer. Only the displayed chain was used for superpositioning the structures in PyMol.
(TIF)

**S7 Fig. Hemagglutination potency of HAdV-D36 and HAdV-D37. A** Depiction of the hemagglutination experiment. Human blood was diluted as indicated in the methods. **B** Quantification of the number of virus particles per cell required for complete agglutination of human red blood cells.
(TIF)

**S8 Fig. Glycan array screening for the HAdV-D37.** A large glycan array containing 498 glycans (see legend on the upper right) was in good overall agreement with previous findings [36]. Numerical scores for the binding intensity are shown as means of fluorescence intensities of duplicate spots at 5 fmol/spot, with error bars representing half of the difference between the two values.
(TIF)

**S9 Fig. Binding of differentially acetylated Neu5Ac variants by HAdV-D36. A** Complex structure of HAdV-D36 with synthetic α-2-*O*-methyl-Neu4,5,9Ac$_3$. Shown is a 2F$_o$-F$_c$ map contoured at 1.5σ (dark blue) and 1σ (light blue), respectively. The additional 9-*O*-acetyl group displays various conformations and does not contribute to the binding. **B** Modelling of additional *O*-acetylations at position 7 and 8 (purple). Modelling was performed in Coot without application of a force field. In agreement with [71], the acetyl groups were added in a sterically favorable position that retains good ligand geometry. Steric clashes are probable in both cases (red discs represent the clashing regions).
(TIF)

**S10 Fig. Comparison of the Neu5Ac and Neu4,5 Ac2 binding between HAdV-D36 (upper panel) and HAdV-D37 (Lower panel).** Coloring of interacting residues according to **Fig 5**. **A-C** Neu4,5Ac$_2$ (green) binding mode observed for the HAdV-D36 FK (gray surface) analogous to **Fig 5A–5B**. **D-F** Neu4,5Ac$_2$ (green) binding mode observed for the HAdV-D37 FK (gray surface). The triangular hydrophobic interaction is not observed in HAdV-D37. Instead,

Neu4,5Ac$_2$ moieties are located in the periphery of the binding cavity. The relative movement between Neu5Ac (yellow, PDB-ID 1UXA) and Neu4, 5Ac2 is less pronounced for HAdV-D37 than for HAdV-D36.
(TIF)

**S11 Fig. Gatekeeping residues of HAdV-D36 and different nidovirus hemagglutinin-esterases.** Superpositioning was done according to the carbohydrate portions using PyMol. **A** Superposition of HAdV-D36 (green), mouse hepatitis virus strain S hemagglutinin-esterase (MHV-S HE[0], cyan, PDB ID 4C7W), and the lectin (yellow) and catalytic (orange) sites of rat coronavirus strain New-Jersey hemagglutinin-esterase (RCoV-NJ HE[0], PDB ID 5JIL) in complex with the non-hydrolysable Neu4,5Ac$_2$ analogue α-2-O-methyl-Neu4,5-di-N-acetylneuraminic acid. **B** Gatekeeping residue of MHV-S HE[0]. **C, D** Gatekeeping residues of the RCoV-NJ HE[0] esterase (orange) and lectin (yellow) binding sites.
(TIF)

**S12 Fig. Quaternary arrangements of HAdV-D36/D37 FK wild type and trimer interface mutants.** All superpositions were performed in PyMol aligning all atoms of the trimer. **A** Superposition of the HAdV-D36 wt FK (gray) with the HAdV-D37 wt FK (teal) in analogy to [Fig 3B](). Below are the separate surface representations of the two wt knobs. **B** Superposition of the HAdV-D36 YGT FK (orange) with the HAdV-D37 VSN FK (dark gray). Below are the separate surface representations of the two mutant knobs. The HAdV-D37 VSN FK shows a narrower central cavity than HAdV-D36 YGT. A clockwise movement of the Y317 side chain in HAdV-D36 YGT distorts the N-acetyl binding cavity (both analogous tyrosine residues shown as sticks). C Superposition of SA complexes of HAdV-D36 wt. The interacting residues were aligned on the sidechain of the gatekeeping residue Y315. The O-acetyl group of Neu4,5Ac$_2$ (green) is in the van-der-Waals range of sidechains N313, Y315, and P323, while the N-acetyl group forms a direct hydrogen bond with N313*. D Modeling of Neu4,5Ac$_2$ (green) on the HAdV-D36/Neu5Ac complex structure. Neu4,5Ac$_2$ was aligned onto Neu5Ac (yellow) in Coot. The relative spacing of T313* and the N-acetyl group prevent the formation of a water-mediated contact. Instead, the N-acetyl group moves towards T313* and forms a direct, long hydrogen bond. Thereby, the N-acetyl function is slightly rotated out of its ideal position and causes a rotation of the whole ligand. As a consequence, the O-acetyl function of a possible Neu4,5Ac$_2$ complex would produce heavy clashes with T313, Y315, and P323 that cannot be overcome by a rotation towards the center. E Superposition of SA complexes of HAdV-D37 wt. T310* is located further away from the sugar's N-acetyl group and allows for the formation of a water-mediated contact in both cases. F Superposition of the Neu5Ac complex structures of HAdV-D36 wt, HAdV-D36 YGT, and HAdV-D37 wt. The need to form a direct contact in HAdV-D36 YGT pulls the sugar towards the gatekeeping residue Y315. Coloring according to C-E. G Superposition of the Neu4,5Ac$_2$ complex structures of HAdV-D36 and D37 wt with the modelled complex of HAdV-D36 YGT. The downward motion and rotation of the sugar in the latter induce heavy clashes. Coloring according to C-E.
(TIF)

**S13 Fig. Comparison of the Neu4,5Ac2 binding modes of HAdV-D37 wt, HAdV-D37 VSN, and HAdV-D36 wt.** Residues at the analogous positions 310 (HAdV-D37) and 313 (HAdV-D36) are displayed as sticks and their contacts with the sugar as black dashed lines. **A** Binding mode of HAdV-D37 wt (pink sticks, light surface). **B** Superposition of the Neu4,5Ac$_2$ complex structured of HAdV-D37 wt and VSN (cyan sticks, dark surface). The triangular hydrophobic contact is restored in the VSN mutant. **C, D** Binding mode of HAdV-D37 VSN. The same panel is displayed twice for reasons of clarity. **E** Superposition of the Neu4,5Ac$_2$

complex structures of HAdV-D37 VSN and HAdV-D36 wt (green sticks, light surface). Despite a relative clockwise and upward positioning of the sugars in HAdV-D36, the length of the triangular hydrophobic contact is very similar. **F** Binding mode of HAdV-D36 wt analogous to **Fig 5D**.
(TIF)

**S14 Fig. Purification of 4-O-Ac-3'SL from echidna milk oligosaccharides (EMOs). A** Bio-Gel P4 Fractionation of EMOs. **B** HPLC of Bio-Gel P4 fraction F6 **C** Re-HPLC of Bio-Gel P4 fraction F6-2. **D** Negative-ion mass spectra of HPLC fraction F6-2. **E** Negative-ion mass spectra of HPLC fraction F6-2a **F** Negative-ion mass spectra of HPLC fraction F6-2b. **G** $^1$H-NMR of HPLC F6-2a.
(TIF)

**S1 Table. List of probes included in microarray screening analysis (Array Sets 32–39); their sequences and fluorescence intensities at 5 fmol per spot of binding with His-tagged fiber knobs of HAdV-D36 and HAdV-D37.**
(PDF)

**S2 Table. Excerpt from the main S1 Table of the fluorescence signals obtained for large arrays of HAdV-D36 and HAdV-D37.**
(PDF)

**S3 Table. Data collection and refinement statistics of selected datasets.**
(PDF)

**S4 Table. Supplementary glycan microarray document based on MIRAGE Glycan Microarray guidelines (doi:10.3762/mirage.3) for the glycan array experiments.**
(PDF)

**S1 Text. Extended Materials and Methods.**
(PDF)

## Acknowledgments

The authors wish to thank Alexandra Thor and Dr Michael Braun (Interfaculty Institute of Biochemistry, University of Tübingen) for providing purified CAR-D1 and CD46-D4, respectively, and Professor Tadasu Urashima (Obihiro University, Japan) for generously supplying valuable echidna milk oligosaccharides for the study. We are also grateful to the staff at the Swiss Light Source (SLS, Villigen, Switzerland) for beam time and beamline assistance. The sequence-defined glycan microarrays contain many saccharides provided by collaborators whom we thank as well as members of the Imperial College London Glycosciences Laboratory for their contribution in the establishment of the NGL-based microarray system.

## Author Contributions

**Conceptualization:** A. Manuel Liaci, Mikael Elofsson, Ten Feizi, Yan Liu, Thilo Stehle, Niklas Arnberg.

**Data curation:** A. Manuel Liaci, Mikael Elofsson, Niklas Arnberg.

**Formal analysis:** A. Manuel Liaci, Naresh Chandra, Sharvani Munender Vodnala, Michael Strebl, Pravin Kumar, Vanessa Pfenning, Paul Bachmann, Rémi Caraballo, Wengang Chai, Emil Johansson, Niklas Arnberg.

**Funding acquisition:** Mikael Elofsson, Ten Feizi, Yan Liu, Thilo Stehle, Niklas Arnberg.

**Investigation:** A. Manuel Liaci, Naresh Chandra, Sharvani Munender Vodnala, Michael Strebl, Pravin Kumar, Vanessa Pfenning, Paul Bachmann, Rémi Caraballo, Wengang Chai, Emil Johansson, Yan Liu, Niklas Arnberg.

**Methodology:** A. Manuel Liaci, Naresh Chandra, Sharvani Munender Vodnala, Michael Strebl, Pravin Kumar, Vanessa Pfenning, Paul Bachmann, Rémi Caraballo, Wengang Chai, Emil Johansson, Niklas Arnberg.

**Project administration:** Mikael Elofsson, Ten Feizi, Yan Liu, Thilo Stehle, Niklas Arnberg.

**Resources:** Niklas Arnberg.

**Supervision:** Mikael Elofsson, Ten Feizi, Thilo Stehle, Niklas Arnberg.

**Validation:** Niklas Arnberg.

**Writing – original draft:** A. Manuel Liaci, Thilo Stehle, Niklas Arnberg.

**Writing – review & editing:** A. Manuel Liaci, Mikael Elofsson, Yan Liu, Thilo Stehle, Niklas Arnberg.

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
