## [Decision Letter · Decision Letter 0]

11 Nov 2024

PPATHOGENS-D-24-02313Extended receptor repertoire of an adenovirus associated with human obesityPLOS Pathogens Dear Dr. Arnberg, Thank you for submitting your manuscript to PLOS Pathogens. After careful consideration, we feel that it has merit but does not fully meet PLOS Pathogens's publication criteria as it currently stands. Therefore, we invite you to submit a revised version of the manuscript that addresses the points raised during the review process. Please submit your revised manuscript within 30 days Jan 10 2025 11:59PM. If you will need more time than this to complete your revisions, please reply to this message or contact the journal office at plospathogens@plos.org. Please include the following items when submitting your revised manuscript:*
A rebuttal letter that responds to each point raised by the editor and reviewer(s). You should upload this letter as a separate file labeled 'Response to Reviewers'. This file does not need to include responses to any formatting updates and technical items listed in the 'Journal Requirements' section below.*
A marked-up copy of your manuscript that highlights changes made to the original version. You should upload this as a separate file labeled 'Revised Manuscript with Track Changes'.*
An unmarked version of your revised paper without tracked changes. You should upload this as a separate file labeled 'Manuscript'. If you would like to make changes to your financial disclosure, competing interests statement, or data availability statement, please make these updates within the submission form at the time of resubmission. Guidelines for resubmitting your figure files are available below the reviewer comments at the end of this letter. We look forward to receiving your revised manuscript. Kind regards, Ekaterina E. HeldweinAcademic EditorPLOS Pathogens Robert KalejtaSection EditorPLOS Pathogens Michael Malim

Editor-in-Chief

PLOS Pathogens

orcid.org/0000-0002-7699-2064 **Journal Requirements:** **Additional Editor Comments (if provided):** Both reviewers were enthusiastic about the reported work and suggested some modifications to the text and figures. Therefore, I am inviting you to revise your manuscript accordingly. While you may wish to consider the experiments suggested by Reviewer 1, these are not required for the current manuscript. However, Reviewer 1 also requested that you thoroughly review the manuscript and figures to address their concerns. Reviewer 2 also had a couple of comments. Please, address all reviewers criticisms to the best of your ability.**Reviewers' Comments:** Reviewer's Responses to Questions

**Part I - Summary**

Reviewer #1: See attached pdf.

Reviewer #2: Manuscript entitled “Extended receptor repertoire of an adenovirus associated with human obesity” by Liaci et al describes the receptor usage of HAdV-D36, which is implicated in the obesity in animals and humans. The manuscript comes from the highly reputed research laboratories in virology and structural biology and reports the results on an important topic of adenoviral entry mechanisms and is well written. Therefore, it is well suited for publication in PLOS Pathogens.

I have a few comments on the scope of the findings and minor suggestions.

1) Authors say in the abstract that the elongated DG loop of the Ad36 fiber knob (FK) alters the known CAR binding interfaces and “heavily clashes” with CAR when superimposed onto a known FK-CAR complex structure (Fig. S4). These statements imply that Ad36-FK is unlikely to bind to CAR. But authors show using cell-based assays that the presence of CAR increases the binding of Ad36 in some of the cell lines tested (Fig. 1C). Is it possible that Ad36 actually binds to sugar moieties (e.g., sialic acids) on glycosylated CAR? Is there a way to prove or disprove such a possibility (e.g., deglycosylation of CAR expressed on CHO cells)?

2) Is there a requirement that Neu5Ac moieties to be the terminal sugars as part of the complex carbohydrates to bind to Ad-FKs. Authors may be able to answer this question from the known organization of sugars from the glycan array they used in the study. Along those lines, can more than two consecutive sugars (as part of the complex carbohydrates) bind to the 2-3 binding sites on a FK?

3) Figure 3. Since the authors are comparing the relative electrostatic potentials of Ad36 and Ad37 FKs to make a point, it would be useful to include a panel showing the electrostatic potential of Ad37-FK in addition to that of Ad36-FK. Moreover, in the reviewer’s opinion, the superposition of FKs (Panel B) may be visualized better as a ribbon representation than as a surface representation.

Minor comment.

Line 33 (Page 2). In the abstract…HAdV-D36 has been causally linked…

consider replacing “causally” with “putatively”

**Part II – Major Issues: Key Experiments Required for Acceptance**

Reviewer #1: See attached pdf.

Reviewer #2: (No Response)

**Part III – Minor Issues: Editorial and Data Presentation Modifications**

Reviewer #1: See attached pdf.

Reviewer #2: (No Response)

PLOS authors have the option to publish the peer review history of their article (what does this mean?). If published, this will include your full peer review and any attached files.

Reviewer #1: No

Reviewer #2: **Yes: **Vijay S. Reddy

---

## [Decision Letter · Decision Letter 1]

7 Jan 2025

Dear Professor Arnberg,

We are pleased to inform you that your manuscript 'Extended receptor repertoire of an adenovirus associated with human obesity' has been provisionally accepted for publication in PLOS Pathogens.

Best regards,

Ekaterina E. Heldwein

Academic Editor

PLOS Pathogens

Robert Kalejta

Section Editor

PLOS Pathogens

Sumita Bhaduri-McIntosh

Editor-in-Chief

PLOS Pathogens

orcid.org/0000-0003-2946-9497

Michael Malim

Editor-in-Chief

PLOS Pathogens

orcid.org/0000-0002-7699-2064

Reviewer Comments (if any, and for reference):

Reviewer's Responses to Questions

**Part I - Summary**

Reviewer #1: The authors have revised the manuscript and supporting materials, including all figures, as requested. In my opinion, the manuscript is now suitable for publication in PLOS Pathogens.

Reviewer #2: Authors have adequately answered all my queries. I don't have further comments or concerns. Accept as is.

**Part II – Major Issues: Key Experiments Required for Acceptance**

Reviewer #1: (No Response)

Reviewer #2: (No Response)

**Part III – Minor Issues: Editorial and Data Presentation Modifications**

Reviewer #1: (No Response)

Reviewer #2: (No Response)

PLOS authors have the option to publish the peer review history of their article (what does this mean?). If published, this will include your full peer review and any attached files.

Reviewer #1: No

Reviewer #2: **Yes: **Vijay S. Reddy

---

## [Editor Report · Acceptance letter]

24 Jan 2025

Dear Professor Arnberg,

We are delighted to inform you that your manuscript, "Extended receptor repertoire of an adenovirus associated with human obesity," has been formally accepted for publication in PLOS Pathogens.

Best regards,

Sumita Bhaduri-McIntosh

Editor-in-Chief

PLOS Pathogens

orcid.org/0000-0003-2946-9497

Michael Malim

Editor-in-Chief

PLOS Pathogens

orcid.org/0000-0002-7699-2064